# Ecosystem Management Policy Implications Based on Tonga Main Tuna Species Catch Data 2002–2018

**Siosaia Vaihola** [1],* **and Stuart Kininmonth** [2]

1   School of Agriculture, Geography, Environment, Ocean and Natural Sciences, University of the South Pacific, Suva, Fiji

2   Heron Island Research Station, Faculty of Science, University of Queensland, Gladstone 4680, Australia; s.kininmonth@uq.edu.au

*   Correspondence: saiavaiholajr@gmail.com or s92608460@student.usp.ac.fj; Tel.: +679-77933126

**Abstract:** Despite the crucial role played by international and regional tuna fisheries in facilitating the successful implementation of the ecosystem approach to fisheries management, there exist disparities in viewpoints among these stakeholders, resulting in gaps between regional fisheries management and local communities. Nevertheless, the Tongan government, under the Ministry of Fisheries, is dedicated to the efficient management of its tuna resources, aiming to establish it as the preferred and optimal approach for ensuring the long-term sustainability of its tuna fisheries and the ecosystem services they provide to the community. Recognizing that an appropriate legal, policy and institutional framework is in place for sustainable management of tuna, the first part of this paper presents a review of current Tonga fisheries laws and policies for its tuna fisheries. This review reflects the implementation of an information-based management framework, namely the Tonga National Tuna Fishery Management and Development Plan. The tuna fisheries in Tonga mainly catch albacore (*Thunnus alalunga*), bigeye (*Thunnus obesus*), skipjack (*Katsuwonus pelamis*), and yellowfin (*Thunnus albacares*) tuna. These tuna species are caught within Tonga's exclusive economic zones and play a crucial role in the country's economy; hence, it is crucial to examine the spatio-temporal distributions of their catch in relation to their environmental conditions. In pursuit of this goal, the tasks of mapping (i) the spatio-temporal distribution of catch landed at ports and (ii) the spatio-temporal of environmental conditions were performed. The study utilizes longline catch per unit effort data spanning from 2002 to 2018 for albacore, bigeye, skipjack, and yellowfin tuna. It also incorporates data on environmental conditions, including sea surface temperature, sea surface chlorophyll, sea surface current, and sea surface salinity. Additionally, the El Nino Southern Oscillation Index is mapped in relation to catch data to examine the potential effects of climate change on the tuna catch. Results show that bigeye, skipjack, and yellowfin CPUE show a central–northernmost distribution and are primarily caught between latitudes 14° S–22° S, while albacore, shows a central–southern distribution. The highest CPUE for all species are in latitudes 15.5° S–22.5° S and longitudes 172.5° W–176.5° W. The data indicate that sea surface current velocities range from −0.03 to 0.04 ms$^{-1}$, sea surface salinity ranges from 34.8 to 35.6 PSU, sea surface chlorophyll concentration varies from 0.03 to 0.1 mg m$^{-3}$, and sea surface temperature fluctuates seasonally, ranging from 18 °C to 30 °C. Mapping also reveals that times of reduced catches in Tonga coincide with periods of moderate to strong El Nino events from 2002 to 2018.

**Keywords:** economically important species; exclusive economic zone; fisheries research; national obligations; tuna fishery management

## 1. Introduction

Tonga and many other Pacific Island Countries (PICs) in the Western and Central Pacific Ocean (WCPO) depend on tuna fisheries for food security, revenue, and social livelihoods [1–4]. Tongan fisheries largely targeted reef and lagoon species up to the

early 1960s, but due to a local population rise this resulted in overfishing of many inshore marine species. As a result, open water fishes, such as tuna, became the targeted species and soon attracted international fleets to Tongan waters [5]. International longline fleets from countries, like Taiwan, Korea, and the US mainly dominated the fishing of tuna until the early 1980s. By 2001, the number of registered, locally based foreign vessels increased to 25, and then to 33 by 2003 [6]. However, a moratorium upon foreign fishing fleets (2004–2011) caused the decline in tuna longline vessels to 3 by 2011 [6]. The moratorium was lifted in 2011 as part of the Tonga's long-term plan to expand its tuna fishery industry, which resulted in the licensing of 19 foreign longline vessels to fish in Tongan waters in 2012 and 2013 (Table 1) [6]. This expansion was based on the following economically important tuna species: *albacore*, *bigeye*, *yellowfin*, and *skipjack* [6].

**Table 1.** Historical landings of catch, CPUE, total number of hooks used, number of longline vessels operated, and total allowable catch for the main tuna species from 2002 to 2018 within the EEZ, longitude 171.31° W–179.10° W, of Tonga.

| Year | Total No. of Hooks | No. of Longline Vessels | | Catch Size (Metric Tons) | | | | CPUE (No. of Fish/100 Hooks/Year) | | | |
|---|---|---|---|---|---|---|---|---|---|---|---|
| | | Domestic | Foreign | Albacore | Bigeye | Skipjack | Yellowfin | Albacore | Bigeye | Skipjack | Yellowfin |
| 2002 | 38,526 | 17 | - | 740 | 124 | 4 | 170 | 30,890 | 5097 | 209 | 6918 |
| 2003 | 46,622 | 23 | - | 489 | 76 | 15 | 240 | 19,164 | 2702 | 754 | 8686 |
| 2004 | 26,348 | 20 | - | 237 | 47 | 3 | 208 | 10,607 | 2120 | 166 | 9215 |
| 2005 | 28,521 | 13 | - | 235 | 78 | 3 | 123 | 10,290 | 3609 | 163 | 5653 |
| 2006 | 33,818 | 11 | - | 383 | 83 | 2 | 176 | 15,835 | 3859 | 101 | 7439 |
| 2007 | 31,347 | 12 | - | 336 | 109 | 1 | 278 | 14,518 | 4967 | 43 | 12,314 |
| 2008 | 22,432 | 9 | - | 227 | 72 | 0 | 248 | 10,355 | 3441 | 19 | 11,118 |
| 2009 | 11,112 | 6 | - | 146 | 33 | 1 | 97 | 7444 | 1776 | 49 | 5308 |
| 2010 | 6927 | 6 | - | 105 | 19 | 1 | 40 | 4348 | 1064 | 35 | 2513 |
| 2011 | 8703 | 3 | - | 88 | 14 | 2 | 142 | 3170 | 824 | 72 | 6960 |
| 2012 | 48,766 | 4 | - | 829 | 126 | 4 | 379 | 19,846 | 2976 | 121 | 11,488 |
| 2013 | 109,494 | 3 | 19 | 1583 | 230 | 9 | 640 | 36,947 | 5477 | 210 | 17,078 |
| 2014 | 31,357 | 4 | 19 | 284 | 40 | 8 | 378 | 8742 | 1484 | 305 | 14,785 |
| 2015 | 45,302 | 4 | 14 | 724 | 129 | 13 | 755 | 19,822 | 4104 | 364 | 23,191 |
| 2016 | 58,498 | 4 | 4 | 1265 | 159 | 31 | 895 | 32,618 | 4457 | 943 | 28,260 |
| 2017 | 55,438 | 6 | 8 | 874 | 129 | 41 | 871 | 23,328 | 3740 | 1290 | 29,104 |
| 2018 | 30,186 | 6 | 4 | 677 | 63 | 12 | 336 | 21,489 | 2486 | 485 | 13,895 |
| Total allowable catches for each species (metric tons) | | | | 2500 | 2000 | Unlimited | 2000 | Manage through WCPFC harvest strategic plan and TMDP | | | |

Note. No domestic and foreign vessels were licensed to fish in the Tonga EEZ in 2002–2003 and 2002–2012, respectively, as indicated with dash marks.

Regionally, management of tuna fisheries in the PICs is at a critical stage [3,4]. Countries in the WCPO continue to struggle with conflicting interests and issues pertaining to tuna management [7]. These issues include differences in access fees by distant water fishing nations (DWFNs) [8] and transshipment measures and harvest control rules for catch limits [8,9]. These complicate attempts to regulate management efforts within PIC's exclusive economic zones (EEZs) [10,11]. Additionally, the regional fisheries management organizations (RFMOs), such the Western and Central Pacific Fisheries Commission (WCPFC) [12–14], have established comprehensive management regimes for high seas stocks. However, these management actions have met with very little success due to the failures of PICs to adequately implement management measures and sustainability improvement processes [15]. In addition, the rise in population growth poses a threat in the form of further overfishing of tuna [16].

Tuna, a valuable marine resource, holds immense significance for both global fisheries and local communities. Managing tuna stocks effectively demands a holistic approach that bridges the gap between regional fisheries management and the perspectives of local stakeholders. This link is especially vital in the WCPO context, where the WCPFC acts as the primary authority overseeing tuna management within the region encompassing Tonga [17]. Moreover, the presence of two sub-regional groups—the Parties to the Nauru Agreement (PNA) and the Smaller Pacific Island States and Territories (SPG)—adds complexity to the management landscape, as their differing interests often result in divergent views [18]. From a broader regional fishery perspective, the involvement of the WCPFC is essential

due to the transboundary nature of tuna stocks. Tuna species, such as *skipjack*, *yellowfin*, and *bigeye*, traverse vast distances within the WCPO, requiring collaborative management efforts [19,20]. The WCPFC serves as a platform where PICs and DWFNs deliberate on strategies to ensure the conservation and sustainable use of these stocks [19].

However, successful tuna management is not confined solely to international discussions; it must be woven into the fabric of local communities that directly rely on these resources [21]. The importance of linking regional management to local perspectives becomes evident when considering the ecological, economic, and cultural significance of tuna to the Pacific Island communities. Therefore, decisions made at the regional level through the WCPFC directly impact the lives of local fishermen, processors, and communities. An effective management approach acknowledges the need to balance conservation goals with the socio-economic wellbeing of these communities [22]. Within the WCPO, the existence of the PNA and SPG introduces additional layers of complexity to tuna management. These sub-regional groups represent the diverse interests of the PICs [23]. This divergence arises from their distinct geographical, economic, and cultural contexts. Such diversity enriches discussions but can also lead to complexities when trying to reach consensus on management measures. It underscores the importance of diplomacy, negotiation, and compromise in the pursuit of effective policies that safeguard tuna stocks and support local communities [23,24].

The most pressing management question is, therefore, whether each nation should individually manage the fish within its EEZ, or whether a regional agency should manage the species at a broader scale. Either way, Tonga is committed to supporting the growth of this important resource at both national and regional levels. This is demonstrated through Tonga's commitment to: (i) implementing and monitoring catch regulations and (ii) limiting the number of locally-based foreign and foreign-licensed longlining vessels allowed to fish in the EEZ of Tonga. These are executed through the implementation of an information-based management plan, namely the Tonga National Tuna Fishery Management and Development Plan (TMDP) that is revised every five years.

However, Tonga, akin to numerous other small island nations, encounters formidable obstacles when it comes to proficiently overseeing highly migratory species, such as tuna. These are attributed to the following reasons. Given Tonga's constrained financial and technical resources, effectively overseeing highly migratory species poses difficulties. This encompasses expenses related to vessel operations, research endeavors, and regulatory enforcement, all of which can place a significant burden on the country's budget, thus, impeding substantial investments in comprehensive management [25]. Tonga's geographic location in the central Pacific means that tuna stocks passing through its waters are part of larger, shared populations that migrate across multiple countries' exclusive economic zones. Managing these species effectively requires cooperation with neighboring nations, which can be complex due to differing interests and capacities [26,27]. Furthermore, Tonga's small domestic market and limited processing capabilities make it heavily reliant on exporting tuna to international markets. Meeting global demand while maintaining sustainable practices can be challenging, particularly with competition from larger, more resource-rich nations [28,29]

Figure 1 illustrates the structural layout of the Tonga tuna fisheries management. This presentation shows that laws and policies serve as the primary framework for overseeing tuna fisheries, while the sustainable management of tuna relies on the enactment of these policies by local tuna stakeholders. It is anticipated that by means of this framework, tuna fisheries will make a significant contribution to the nation's economy and the wellbeing of the entire community. Tonga also believes that adopting an ecosystem-based approach to fisheries management is the most favorable and optimal strategy for ensuring the long-term sustainability of fisheries and the valuable ecosystem services they provide to society. This is why the country incorporates scientific studies to bolster fisheries management and sustainability efforts.

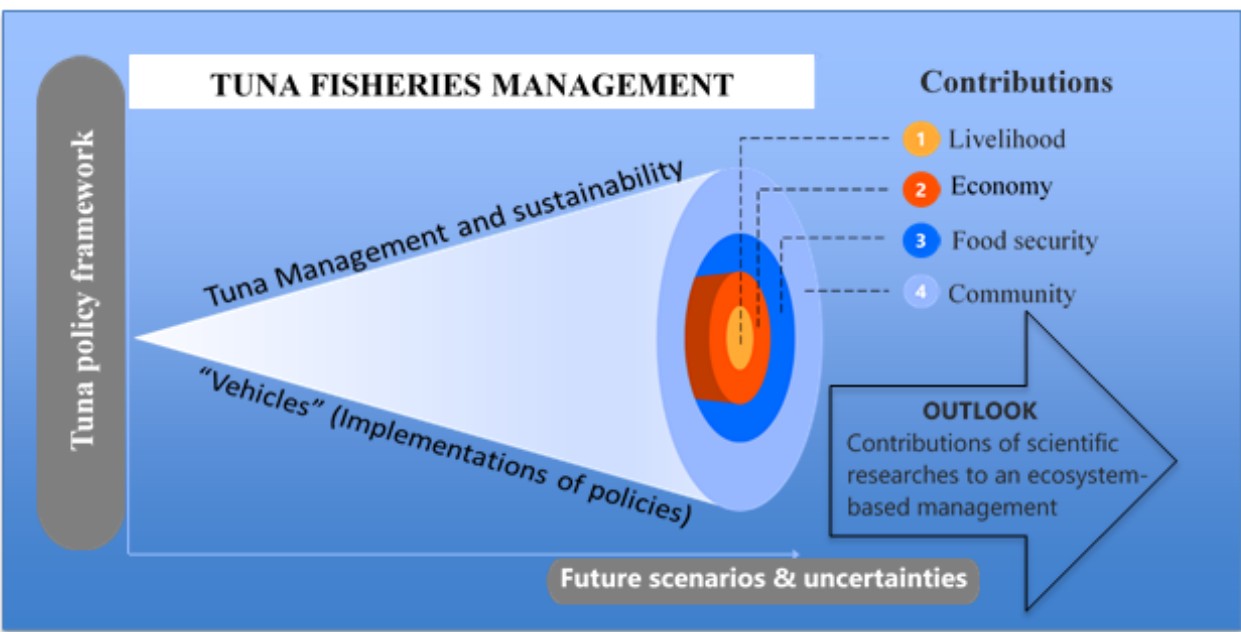

**Figure 1.** Theoretical framework of social and economic interactions between national and local fisheries planners for optimal management of tuna resources and benefits in Tonga. Source: Summarized and modified from Bell et al. (2009) [2] and Tonga's TMDP and Implementation Schedule [6].

## 2. Tonga's Tuna Fisheries and Management Strategies

Tonga is situated within the geographical confines of the WCPO, and its Ministry of Fisheries manages its EEZ with the aim of sustaining the viability of its tuna fisheries. This is achieved through the implementation of its management policies. The predominant tuna species of economic significance in Tonga include *albacore*, *bigeye*, *skipjack*, and *yellowfin*, collectively contributing to over 95% of the total annual catch and economic value of the tuna fisheries [6]. Commercial tuna fishing in Tonga primarily employs the longline method, which was initiated in the 1970s, alongside the occasional use of pole and line techniques [6]. In this section, we delve into Tonga's approaches to promoting the sustainable management of its tuna resources. The initial strategy involves the country's commitment to tuna management, evident in its utilization of information-centric policies that guide and direct fishermen. Moreover, these guidelines delineate regulations aimed at enhancing both profitability and sustainability through effective governance. This governance encompasses the implementation and oversight of catch regulations, economic and livelihood contributions, fleet size control management, and fishing activities within its EEZ.

### 2.1. National Obligations: An Information-Based Management

Although marine resources are freely accessible in Tonga, the Ministry of Fisheries stipulates the operational conditions for tuna fishing within its EEZ [6]. The conditions for the effective management and development of Tonga's tuna fishery are detailed in two primary information-based resources: the TMDP 2018–2022 and the Implementation Schedule [6]. These resources align with Tonga's Fisheries Management Act 2002 and the Tonga Strategic Development Framework II [6]. The creation of these resources involved thorough consultation and engagement with stakeholders. Functioning as high-level policy documents, they offer guidance for the management and growth of the tuna fishery, necessitating the full cooperation of Tonga's tuna fishing industry. The TMDP outlines government goals, strategies for enforcing vessel licensing and compliance, the current state of tuna, and catch limit guidelines. In contrast, the Implementation Schedule establishes strategic directions for executing management actions, focusing on information management, administration, and legal standards within the fishery. Both the TMDP and the Implementation Schedule provide clear directives for entities accessing tuna resources, ensuring that allocations

for food security, livelihoods, and economic growth operate sustainably and efficiently, as illustrated in the theoretical framework flow diagram (Figure 1). These national planners draw from a number of necessary reports and policy documents developed by the Tonga Ministry of Fisheries and regional fishing agencies, such as the WCPFC [6]. Incorporated documents include regional and international arrangements and treaties designed to sustain catches and share benefits within the context of an ecosystem approach to fisheries management [30–32].

## 2.2. Improving Profitability and Sustainability through Governance

In addition to the above national obligations and laws, there are ministerial regulations that govern fishing activities and marine resources management in Tonga. The regulations were clustered in four themes, namely governance in implementing and monitoring catch regulations, contribution to economy and livelihood, governance of fleet size control, and fishing within the EEZ.

### 2.2.1. Governance in Implementing and Monitoring Catch Regulations

One of the ways Tonga cooperates with other PICs in managing tuna resources is its comprehensive sampling of catch landings at designated authorized ports. This is pursuant to the TMDP 2018–2022: "All licensed longline fishing vessels shall offload all catches (100%) in the authorized ports of Tonga". The policy guarantees Tonga's commitment towards the WCPFC's management and conservation strategies for tuna in the PICs. In line with this, Tonga specifically enacts two management steps: first, it rigorously tracks the capture of identified vulnerable species and, second, it assesses the combined catch amounts against yearly quotas for each species [3,4]. At the port, the state inspector verifies that all vessels' relevant identification documentation is true and correct, cross-references authorization for fishing and related fishing activities, and confirms that all fishing gear and devices on-board conform with catch regulations for the species being harvested [6]. In addition, through this comprehensive sampling, Tonga cooperates with the Secretariat of the Pacific Community (SPC) and WCPFC's Offshore Fisheries Program to facilitate sustainable harvesting of tuna. For example, these regional organizations provide species stock status that recommend allowing no catch of Pacific Bluefin based on its fully exploited status in the WCPO [14] and encourage more catch of *skipjack* based on low current catch rates and abundant stock status. Regardless, Tonga's highest catches in 2013 of 2463 mt, and in 2016 of 2350 mt, were insignificant [6] compared to other Pacific nations, as can be seen in the catch records of recent years (Table 1).

A major challenge facing tuna fisheries amongst PICs is the declining fish stocks due to juvenile bycatch by the purse seine fishery using floating objects and fish aggregating devices [3,4]. Tonga is committed to comply with bycatch and non-target species catch regulations, hence, the 100% check of catch landing at authorized ports [6]. Currently, Tonga has no access agreements with DWFN with the exception of the Multilateral Treaty of Fisheries with the United States, which allows US purse seiners to fish within Tongan waters. However, there have been very few US purse seine fishing operations in Tongan waters, due to the low productivity of the EEZ zone as compared to the equatorial belt [28]. Total allowable catch (TAC) for the main commercial tuna species in Tonga's waters are 2500 mt for *albacore*, 2000 mt *bigeye* and *yellowfin*, and no limit for *skipjack* (Table 1) due to its sustained high recruitment rate and abundance in the WCPO [28]. These management limits are set based on and proportionally consistent with TAC recommendations by WCPFC [6].

### 2.2.2. Contribution to Economy and Livelihood

Tuna fisheries have been identified as one of Tonga's most important natural resources [6,15]. The tuna industry in the WCPO is the largest in the world, with annual catches exceeding 2 million metric tons (mt), approximately 50% of the global tuna catch [15]. The largest portion of the catch is taken within the EEZ of the PICs [33,34]. In 2014, PICs

generated approximately USD 820 million from total fishery exports and USD 349 million from total foreign fishing access [35]. Similarly, tuna production is the largest commercial fishery in Tonga, which is estimated at 2000 mt per year (approximately 17% of 30% of Tonga marine resources-related benefits) [6]. Most catches are given by locally based foreign vessels (mainly from Taiwan, the Republic of Korea, and China) that fish in Tonga under a framework of national and regional agreements. These benefits come mainly from foreign fishing vessels' access fees (and related charges) and revenues from domestic and international marketing. The tuna industry also brings other benefits, such as sport fishing, good nutrition through fish protein, and subsistence and artisanal fisheries [2,6]. These benefits also culturally shape and exhibit socially positive effects on livelihood [36]. However, in recent years the tuna fisheries in Tonga have been challenged with rising fuel prices, a decline in tuna prices in both local and international markets, low catch rates, and general economic pressures [6]. Consequently, domestic operators struggle to remain viable despite the technical and policy support provided by the government and international donor agencies [6]. Documentations of national aspirations and strategies exist that attempt to redress and maximize the social and economic benefits of tuna resources, and these will be examined later in this review. In addition, this work reflects how Tonga wishes to develop its tuna resources for the benefits of its people. This highlights the fact that the government realizes that management of tuna resources is a national responsibility [37] and aspires to cooperate undauntedly with other Pacific Island states to overcome economic, social, and climatic challenges.

### 2.2.3. Governance of Fleet Size Control

Longline is the main commercial fishing method used in Tonga. Current mandates limit the number of domestic, locally based foreign, and foreign-licensed longlining vessels to 50, with no more than 10 foreign vessels allowed to fish in the EEZ of Tonga at any given time [6]. Domestic and locally based foreign vessels have license preferences over foreign vessels. The extent of priority is so great that foreign vessels shall be phased out (fishing vessels > 50) when new local vessels apply for licenses. The 2004–2011 moratorium resulted in the decline in longlining vessels from 20 in 2004 to only 3 in 2011. The lifting of the moratorium in 2011 saw the increase in both the number of longline vessels and tuna catch in the subsequent years (Table 1). The number of licensed vessels increased from 3 in 2011 to 23 in 2012, and catch estimates of primary species in 2013 totaled 2463 mt (Table 1), which is over a 40% increase that of the previous year [5]. The purse seine fishery is limited to 150–250 fishing days per fishing vessel per year, which is in line with the WCPFC Vessel Day Scheme (VDS) regulation for purse seiners [3,4].

### 2.2.4. Fishing within the EEZ

Global recognition of a 200-nautical mile EEZ around coastal nations allows PICs to claim vast amounts of maritime resources. As such, it is possible for a domestic fishery to grow if it sustainably utilizes the portion of a regional population that persists within its EEZ [15]. Moreover, on global scale, the PICs' EEZs hold the largest tuna resources and provide some 65–75% of the WCPO's tuna catch [38]. On another note, to date, Pacific Island states have been unable to effectively patrol their EEZs against distant water illegal fishers due to the vast area that requires ongoing observation coupled with a lack of financial, technical, and scientific expertise. Furthermore, regional and national fishing management organizations have barely slowed the decline in key tuna species due to exploitation, unreported fishing, and product distribution [26]. Even so, Tonga is committed to protecting its EEZ by improving its own patrol capabilities and practicing international tuna resource management regimes [6].

Figure 2 shows the geographical boundaries of the EEZ of Tonga which spreads across $14.15^\circ$ S–$22.22^\circ$ S and $171.31^\circ$ W–$179.10^\circ$ W, covering an area of about $596,000$ km$^2$ [39]. This EEZ envelops the northern end of the Tonga Trench, Tonga Ridge, Tofua Arc Volcanic Front, northern end of the Tonga Kermadec Arc, and the westward region of the Lau

Basin [40]. There are two geologically parallel chains of volcanic seamounts along the Tonga Ridge and the famous seamount of Capricorn 193 km east of Vava'u island. These geologically bathypelagic features are part of the island nation's fishing ground. *Albacore* and *yellowfin* dominated Tonga's annual catch from 2002 to 2018 over *bigeye* and *skipjack* (Table 1). In addition, *bigeye* (Figure S1), *skipjack* (Figure S2), and *yellowfin* (Figure S3) catches were primarily higher in the central and southern quadrants of the EEZ, while *albacore* (Figure 3) catches were relatively higher in the northern portion of Tonga's EEZ. Overall, catch value (as indicated by number of metric tons) is higher in the area bounded by 15.5° S–22.5° S and 172.5° W–176.5° W, i.e., central to the northern part of the EEZ.

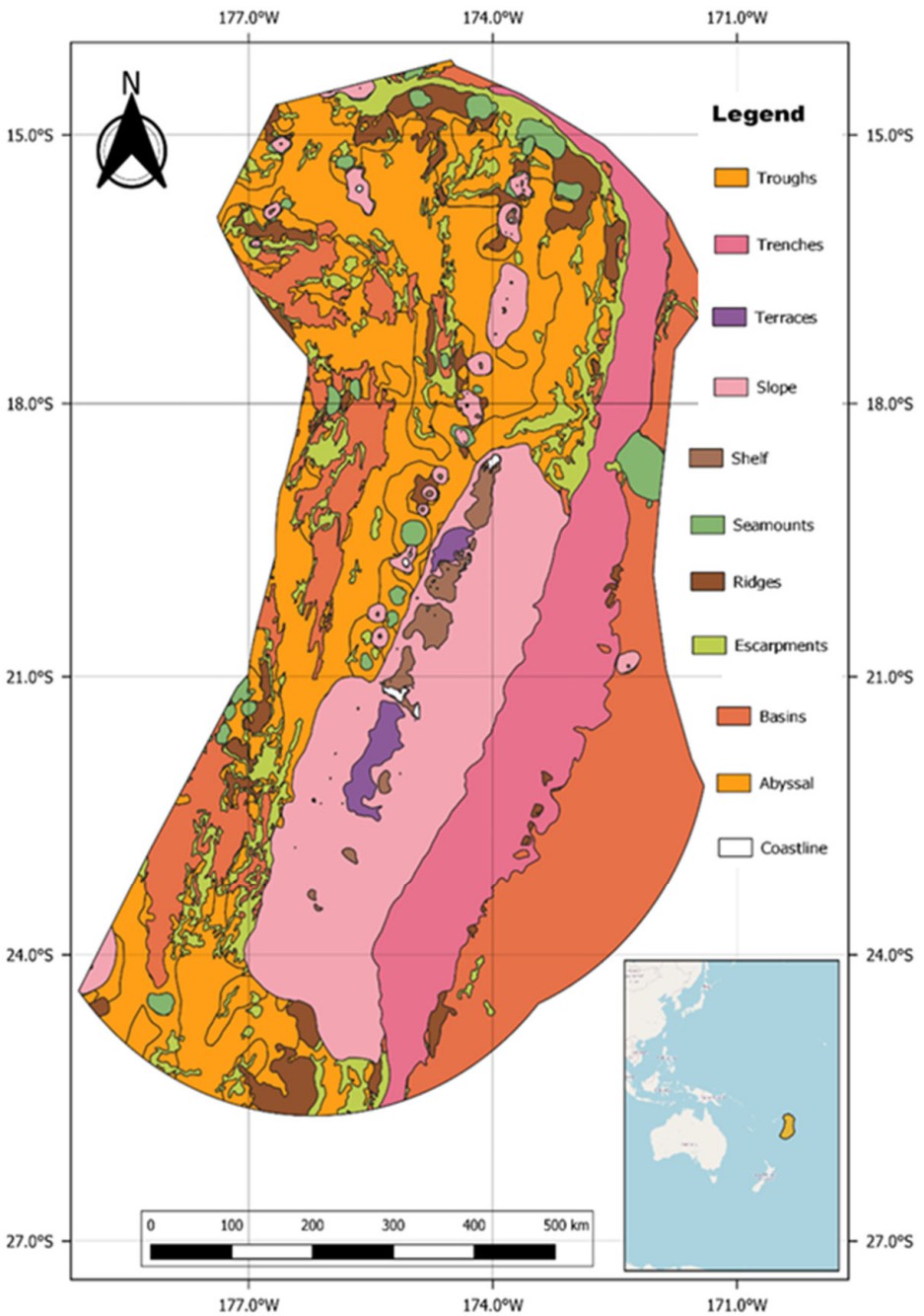

**Figure 2.** The fishing ground (EEZ) of Tonga, which envelops areas with troughs, trench, terraces, slope, shelf, seamounts, ridges, escarpments, basins, and abyssal regions.

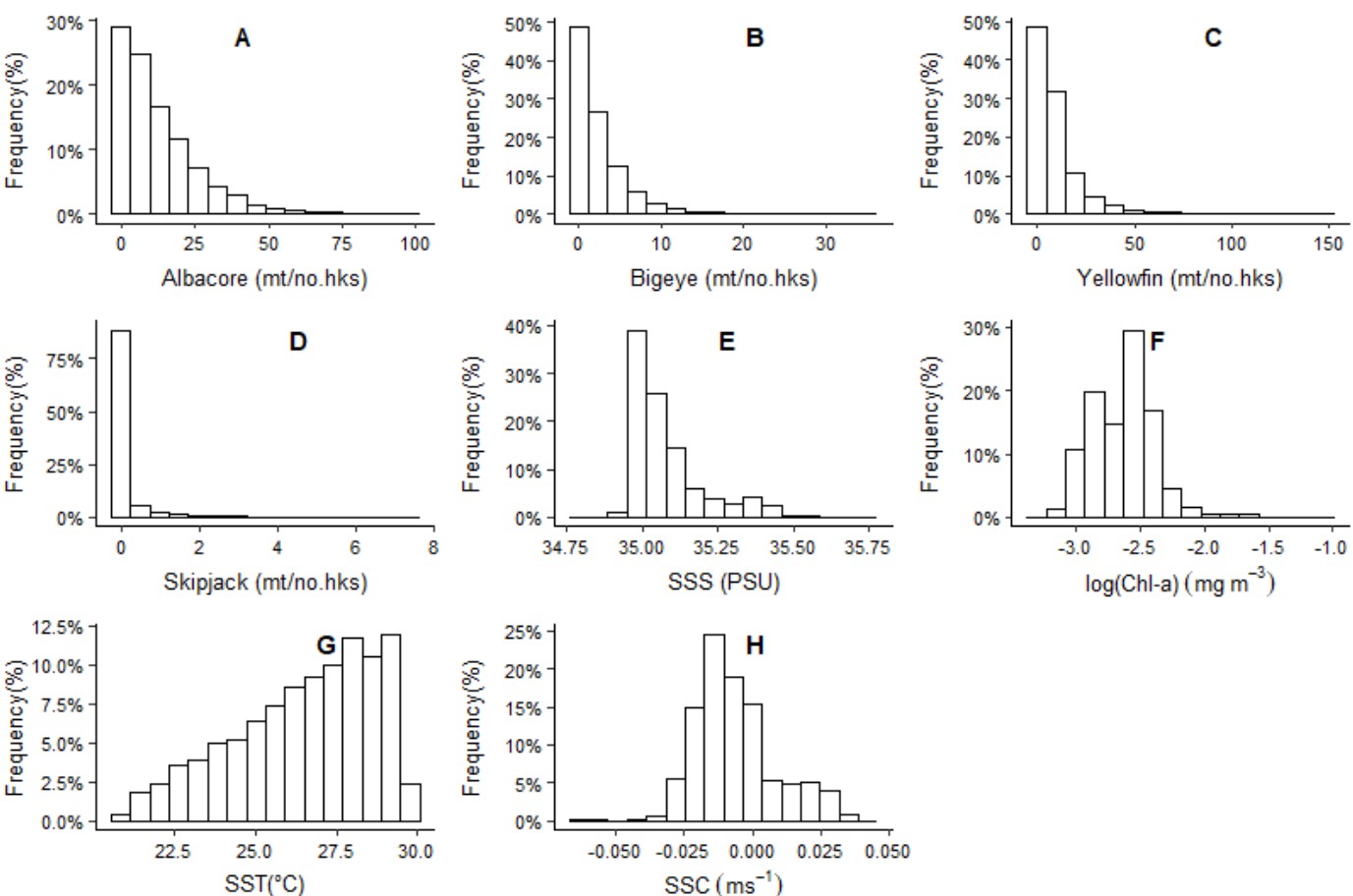

**Figure 3.** Histograms of the spread of CPUE of tuna ((**A**) *albacore*, (**B**) *bigeye*, (**C**) *yellowfin*, (**D**), and *skipjack*) and environmental data (**E**) SSS, (**F**) Chl-*a*, (**G**) SST, and (**H**) SSC) for the period 2002–2018, of the Tonga tuna fisheries and the NASA ECCO-2 Global Circulation Model and AVHRR [41], respectively, taken within the EEZ of Tonga. SSC = sea surface current, SSS = sea surface salinity, and SST = sea surface temperature and chlorophyll concentration.

## 3. Methods

### 3.1. Fishery Data

Data collected between January 2002 and December 2018 for *albacore*, *bigeye*, *skipjack*, and *yellowfin* tuna catches were utilized to create distribution maps of the tuna species within Tonga's EEZ. The catch and effort information was gathered by Tonga's longline fishery and generously provided by the Tonga Ministry of Fishery and the South Pacific Community (SPC) Office located in New Caledonia. Rigorous checks were conducted on the entirety of the fish catch by a minimum of two fisheries offices to ensure compliance [6]. This fishery data was organized into a spatial grid with 1° resolution and included details, such as daily fishing positions (latitude and longitude), fishing effort (expressed metric tons per number of hooks, mt/no.hks), fishing date (recorded by day, month, and year), and catch (reported in numbers).

### 3.2. Environmental Data

Four environmental factors were used to demonstrate the distribution of the four tuna species. The environmental factors included were sea surface current (SSC), sea surface salinity (SSS), sea surface temperature (SST), and chlorophyll-a concentration (Chl-*a*). The SSC and SSS are the monthly/0.25 degree$^2$ of the NASA ECCO-2 Global Circulation Model (ec-co2.jpl.nasa.gov, accessed on 14 May 2022) and SST and Chl-*a* are the daily/0.25 degree$^2$ of the Advanced Very High Resolution Radiometer (AVHRR) sensor

on board the National Oceanic and Atmospheric Administration (NOAA) satellites [41]. We used the *ggplot2* package [42] in the R-software [43] to generate our visualization and the *dplyr* and *tidyverse* packages [44] for data manipulation and analysis. The utilities, *geom_raster* and *geom_point* of *ggplot2*, were used for the overlaying of the environmental and fishery data monthly composites.

### 3.3. Index for Spatio-Temporal Distribution Mapping

For our study, we used catch per unit effort (CPUE), which is calculated as the weight of the catch in metric tons divided by the number of hooks deployed per fishing record, providing a standardized measure of fishing efficiency and effort in capturing the target species [39,40]. The CPUE data were aggregated into monthly and annual resolved datasets to match the temporal scales of the predictor variables in Microsoft Excel [45]. The distributions of the CPUEs of *albacore*, *bigeye*, *yellowfin*, and *skipjack* are shown in Figure 3A–D, respectively. Among the four tuna species, the CPUE values ranged from 0 to approximately 150 mt/no.hks, with the majority of catch encounters falling within the 0 to 75 mt/no.hks range. The distributions of the selected environmental variables are shown in Figure 3E–H, respectively. The SSS values fall within the range of 34.8 to 35.6 PSU, SST values range from 18 to 30 °C, SSC values vary between $-0.03$ to $0.04$ ms$^{-1}$, and Chl-*a* values are within the range of 0.03 to 0.1 mg m$^{-3}$. We employed these indices to superimpose the monthly patterns of CPUE for the four tuna species spanning from 2002 to 2018, along with the selected environmental factors during the same timeframe.

## 4. Results

### 4.1. Spatio-Temporal Distribution of Four Main Tuna Species 2002–2018

Figure 4 shows the monthly catch (in CPUE) distribution of *albacore* (top left), *bigeye* (top right), *skipjack* (bottom left), and *yellowfin* (bottom right) tuna spanning the period from 2002 to 2018. The monthly trend indicates that *albacore* had its peak catch between May and August, *bigeye* between April and July, *skipjack* from June to August, and *yellowfin* from December through February. These periods coincide with the summer season in Tonga, which spans from November to March, and the winter season, from April to October. The figure displays the median, represented by the mark inside the quartiles, and the relative density, indicated by the dimensions (width and height) of the plot, for the data points of each species. *Albacore* and *yellowfin* demonstrate the most catches, followed by *bigeye* and *skipjack* in terms of catch volume.

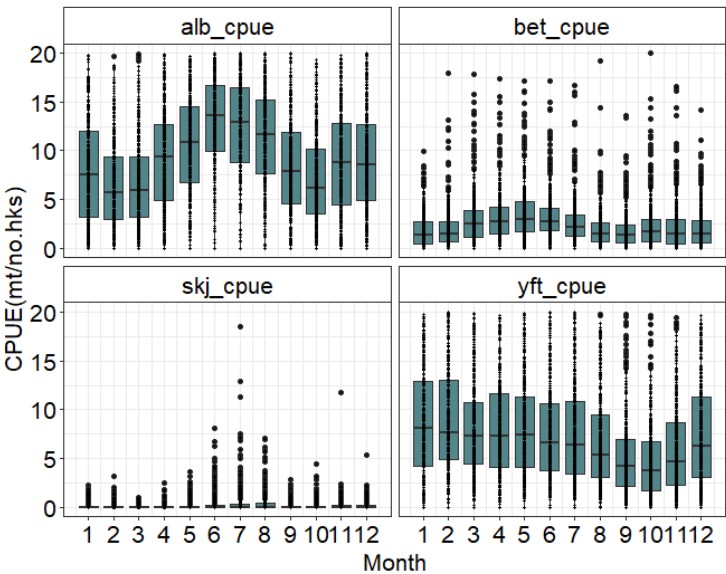

**Figure 4.** Historical monthly catch trends (CPUE) of the four main tuna species in years 2002 to 2018.

Figure 5 shows an overview of the spatio-temporal distribution of the CPUE of *albacore*, *bigeye* (Figure S1), *skipjack* (Figure S2), and *yellowfin* (Figure S3) across the EEZ of Tonga over the time period of 2002–2018. The figures show the catch in CPUE (1° spatial grid) plotted against latitudes and longitudes. Generally, the highest CPUE values are in latitude range 15.5° S–22.5° S and longitude range 172.5° W–176.5° W of the EEZ for all species. The distribution of CPUE suggests that most of the fishing effort of the tuna longliners was concentrated in these geographical ranges. These are evidently the areas most occupied by fishing fleets.

*Bigeye*, *skipjack*, and *yellowfin* CPUE show central–northernmost distribution, and they were primarily caught between latitudes 14 S and 22 S. *Albacore*, on the other hand, thrives in temperate to subtropical regions and displays its highest catch rates in the central and southern parts of the EEZ. The high CPUE for all these species occurred during two periods: from 2002 to 2008 and from 2012 to 2018. Conversely, the lowest CPUE was recorded between 2009 and 2011. It can also be seen that high tuna CPUE values were observed during winter (April–October) and relatively low encounters were observed during summer (November–March). Additionally, the figures illustrate noticeable seasonal variations in the CPUE of the four tuna species on a broad scale, with a notable increase in their CPUE, particularly in recent years.

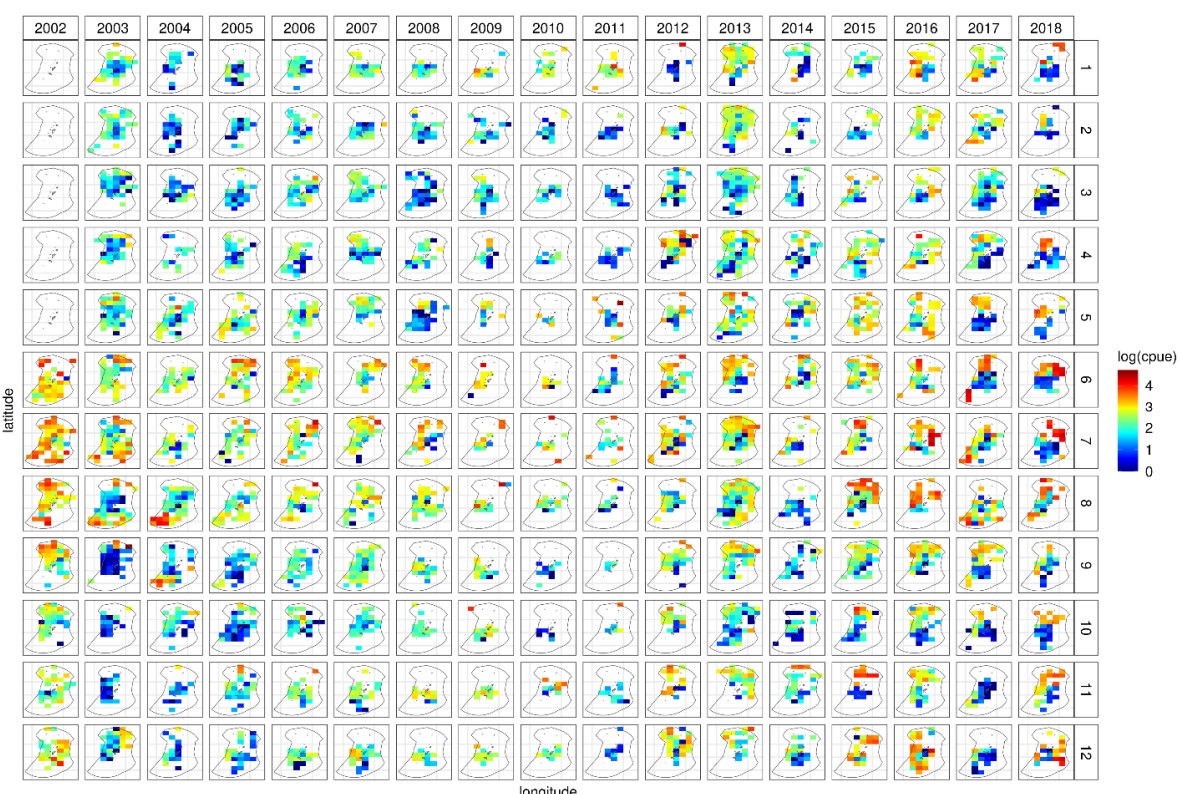

**Figure 5.** Spatio-temporal pattern in the distribution of albacore tuna within the EEZ, longitude 14.5° S–20.22° S, longitude 171.31° W–179.10° W, of Tonga over the time period 2002–2018.

## 4.2. Tuna Habitats in Relation to Biological and Physical Oceanic Conditions

Figure 6 shows an overview of the monthly distributions of SSC (zonal current, uvel in ms$^{-1}$, Figure 6A), SSS (in PSU, Figure 6B), SSS (in °C, Figure 6C), and chlorophyll (Chl-*a* in mg m$^{-3}$, Figure 6D) from 2002 to 2018. It can be seen that the distribution of the zonal current and sea surface salinity are homogeneous throughout the year (Figure 6A,B). Sea surface current is distributed in a range of −0.03 to 0.04 ms$^{-1}$, and sea surface salinity is distributed in a range of 34.8 to 35.6 PSU, respectively. However, a relatively high sea surface current is shown at high latitudes between 22° S–25° S. For sea

surface temperature and Chl-*a*, their distribution is not homogeneous in the EEZ of Tonga. SST has a pronounced seasonal variability with the value ranges from 18 °C to 30 °C (Figure 6C). From December to August, the temperature is relatively cooler, between 20 °C and 28 °C. From September to November the temperature increases to a range between 22 °C and 30 °C. The distribution of Chl-*a* concentration also suggests seasonal variations (Figure 6D). The value of Chl-*a* concentration is at a range of 0.03 to 0.1 mg m$^{-3}$. Generally, higher Chl-*a* concentrations occurred in the central–southern part of the EEZ during winter (from June to September) and lower concentrations occurred in the central–northern part of the EEZ during summer (from October to May).

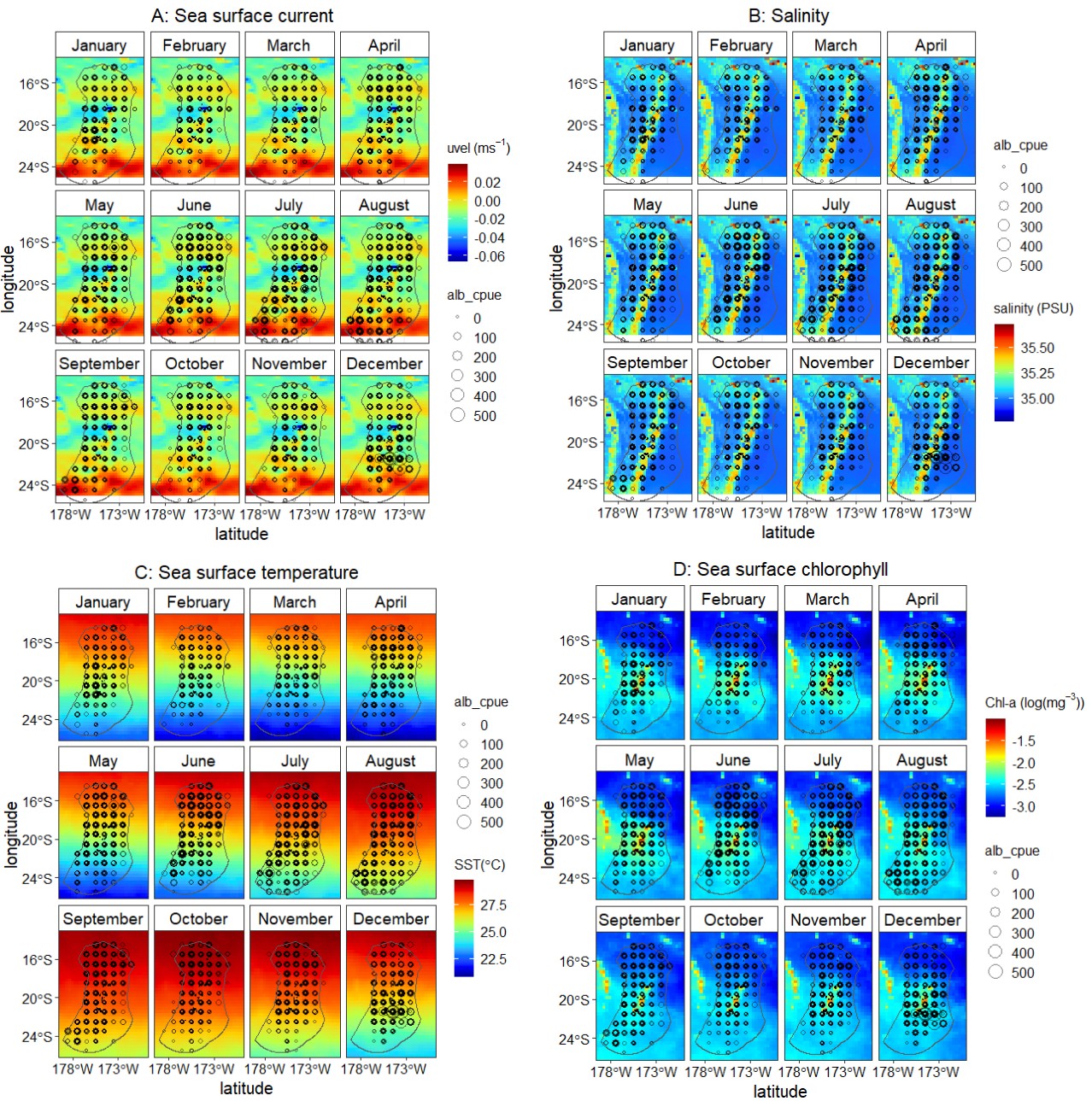

**Figure 6.** The monthly *albacore* CPUE and monthly environmental conditions distribution for (**A**) sea surface current, (**B**) sea surface salinity, (**C**) SST, and (**D**) Chl-*a* within the EEZ, longitude 14.5° S–20.22° S, longitude 171.31° W–179.10° W of Tonga.

### 4.3. Tuna and Climate Variability

Figure 7 displays the annual catch patterns for the four tuna species and the SOI from 2002 to 2018. Additionally, it includes linear forecasts for catch and SOI, as well as a three-month moving average of catch. It can be seen that 2004/2005, 2009–2011, and 2014 are periods with low catches. The shift is precipitated by positive anomalies in the sea surface temperature, which show moderate to strong Eel Nino events in the time period of 1997/98 (strong), 2003/04 (moderate) [46], 2009/10 (moderate to strong) [47], and 2014–2016 (moderate to strong) [48]. These correspond to periods of low catches in Tonga. Our goal is to use this corresponding pattern in catch and El Nino events (indicated by the SOI Index) as a basis for further research.

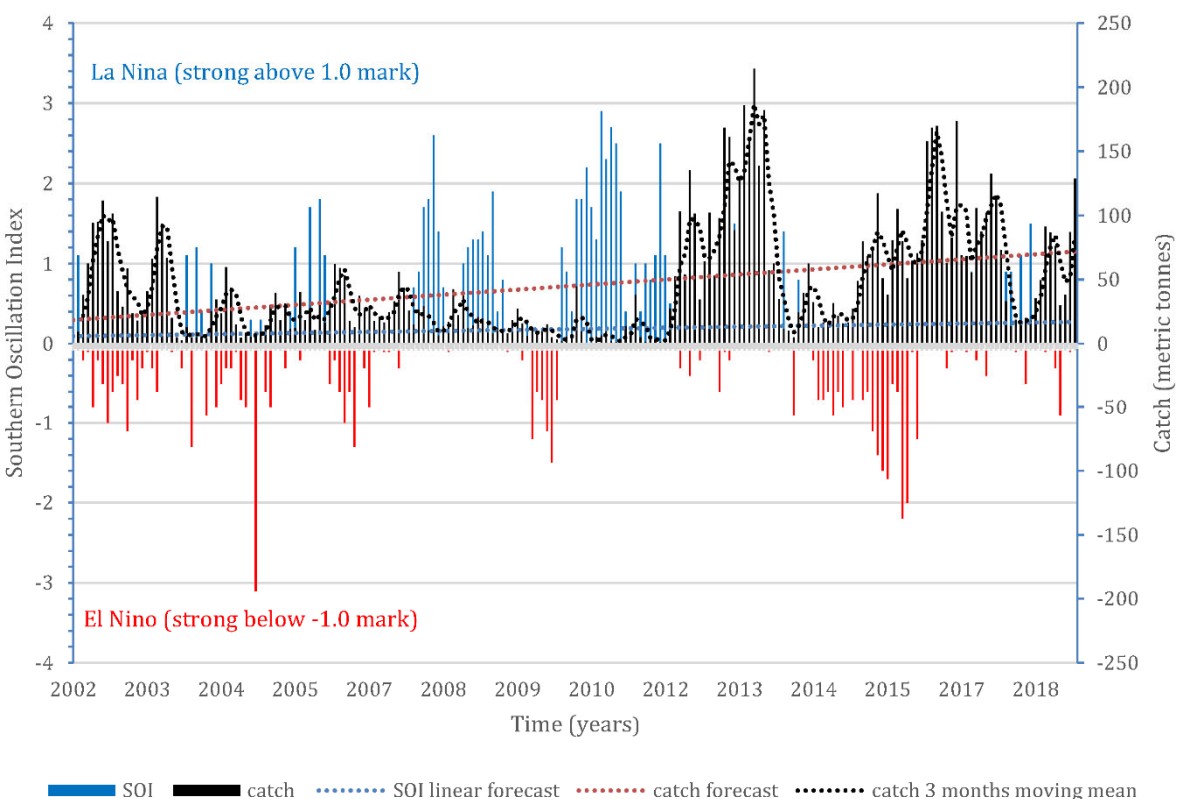

**Figure 7.** SOI and tuna catch patterns of 2002 to 2018 with linear forecasts of catch and SOI, and 3 months moving mean of catch.

### 5. Discussion

Considering the immense value of tuna on both the global and local scales, effective tuna management plays a vital role in ensuring the sustainability and prosperity of tuna resources in the future [49–51]. Achieving effective tuna management extends beyond international discussions; it should be integrated into the everyday life of local communities [50–52] that have a direct dependence on these resources. Hence, effectively overseeing tuna stocks necessitates a collaborative endeavor involving both regional fisheries management and local communities [52]. Tonga is dedicated to fostering the development of this significant resource on both a national and regional scale [6]. This paper outlines the structure through which Tonga observes and strengthens its dedication to its tuna fisheries. This dedication is demonstrated by the administration of pertinent policies and regulations. Tonga's commitment to effective governance is showcased by the following: (i) Execution and supervision of catch regulations and (ii) restriction of the count of foreign-based and foreign-licensed longlining vessels permitted to operate within Tonga's EEZ. These commitments are put into action through the adoption of an information-driven management strategy, specifically the TMDP, which undergoes revision every five years.

This paper underscores Tonga's capacity to promote tuna management on both a national and regional scale, serving as a solid foundation for the sustainability of tuna species and the attainment of maximum benefits.

The CPUE distributions, as depicted in Figures S1–S3 and Figure 5, indicate that in this region, tuna longliners experienced significantly very low tuna encounters compared to the central portion of Tonga's EEZ. We ascribed this distribution of habitat suitability to two primary factors: (i) productive waters in deeper areas are dispersed across larger vertical spans, whereas in shallower areas, these productive waters are concentrated, making them more and quickly accessible for tuna, resulting in higher tuna presence [53,54], and (ii) bathymetry is associated with both fleet operations and fish habitats, serving as a rough indicator of the distance from ports [55,56]. The distance from Nuku'alofa, the capital of Tonga, to the regions with high tuna presence in the central part of the EEZ, falls within the range of 50 to 100 km, while the distance to the deepest areas spans from 150 to 400 km. As a result, it is possible that fishing vessels favored the central area because it offered a shorter distance to ports, thereby reducing operational expenses.

The research conducted by Arrizabalaga et al. (2015) [57] regarding global tuna catches revealed that *albacore* tuna has the highest CPUE and is predominantly caught within the latitude range of 20° South to 40° North. A relatively high CPUE for *bigeye* tuna is observed within the 0° to 40° latitude zone in both hemispheres. *Yellowfin* tuna tend to be located around the equator, while *skipjack* tuna exhibits its highest CPUE in the vicinity of 0° to 20° in each hemisphere. Hu et al. (2018) [56] was discovered that *bigeye* tuna are more frequently caught in equatorial waters that are situated farther away from the coastline and where the hypoxic layer extends to greater depths, as compared to *yellowfin*, *albacore*, and *skipjack* tuna.

By identifying high-density areas, tuna fisheries can concentrate their operations, leading to more efficient and sustainable practices. Conversely, avoiding areas with low CPUE can minimize incidental catch of non-target species, contributing to ecosystem preservation [58,59]. Knowing the spatial distribution of tuna CPUE allows Tonga's fisheries to focus their efforts on areas with higher fish abundance, optimizing catch rates while minimizing operational costs and environmental harm associated with extensive searching. Furthermore, understanding the temporal distribution of tuna CPUE is vital in Tonga's case to ensure proper timing of fishing activities. By identifying seasonal patterns in tuna movements and aggregations, fisheries can align their operations with peak abundance periods, thus, maximizing catches without overexploiting the resource.

Comprehending the spatio-temporal distribution of tuna in relation to their environmental conditions [59] is particularly crucial for Tonga's tuna fisheries given the small size of its EEZ. As a small island nation heavily dependent on tuna resources [6,26] sustainable management of its fisheries is paramount for economic growth, food security, and conservation of marine ecosystems. Our results show that the monthly distributions of SSC and SSS (Figure 5A,B, respectively) are homogeneous throughout the year. The SSC varies within a range of $-0.03$ to $0.04$ ms$^{-1}$, while the SSS ranges between 34.8 to 35.6 PSU. Arrizabalaga et al. (2015) [57] found that *albacore* and *skipjack* showed clearly defined preferred salinity ranges, at around 36–37 PSU, while *bigeye* and *yellowfin* tuna showed a less clearly defined preference of lower salinity of between 34–35 PSU. Nevertheless, *albacore*, *bigeye*, *skipjack*, and *yellowfin* tuna continue to inhabit the waters of Tonga, likely due to other factors influencing their migration, such as their preference for water temperatures. Our distribution maps (Figure 6A) indicate elevated SSC values in high latitudes (between 22° S–25° S), which can be attributed to the strong westerly winds that dominate the high latitudes of the southern hemisphere. These prevailing westerly winds propel surface waters eastward, leading to the enhancement of sea surface currents in those particular areas [60,61]. The spatio-temporal pattern observed in SST and Chl-a data (as shown in Figures 6C and 6D, respectively) indicates that their monthly fluctuations are not uniform but exhibit signs of seasonal variations. The values range from 18 to 30 °C for SST and from 0.03 to 0.1 mg m$^{-3}$ for Chl-*a*. Between December and August, the temperature falls

within the cooler range of 20 °C to 28 °C and, from September to November, the temperature rises, ranging from 22 °C to 30 °C. This indicates seasonal variation in their distributions, which possibly corresponds to seasonal displacement of the water temperature along low–high latitudes [62].

Chl-*a* exhibits seasonal variability, with a higher Chl-*a* concentration occurring in the central–southern part of the EEZ during winter (from June to September) and a lower concentration in the central–northern part of the EEZ during summer (from October to May). Studies have shown that regions with limited catch occurrences are located in the offshore waters because of the low concentration of Chl-*a* in these areas [63,64]. The presence of Chl-*a* serves as an indicator of regions where there is an aggregation of small pelagic fish for feeding purposes [65]. Therefore, it is essential to maintain an adequate level of Chl-*a* concentration to support the availability of food and, consequently, to increase the presence of fish in the area. According to Atkinson et al. (2001) [66], to sustain a viable commercial fishery, a Chl-*a* concentration of at least 0.2 mg m$^{-3}$ is necessary. This is notably higher than the Chl-*a* concentration within the EEZ of Tonga, which ranges from 0.03 to 0.1 mg m$^{-3}$, which is only half of that reported (0.2 mgm$^{-3}$). This is likely because there is no direct nutrient input from land sources and coastal upwellings [67], which typically promote increased primary productivity [56,68].

While there have been no scientific investigations specifically focusing on the habitat preferences of tuna in Tonga, the presence of areas with relatively high CPUE of tuna species may be linked to the oceanographic conditions associated with the bathypelagic features found within Tonga's EEZ. As illustrated by the spatio-temporal patterns in CPUE (as shown in Figure 4), tuna longline vessels encountered regions with a high number of tuna in the central sector of the EEZ. As mentioned earlier, this central area within the EEZ encompasses a portion of the Tonga Ridge, the Tofua Arc Volcanic Front, the northern segment of the Tonga Kermadec Arc, the western expanse of the Lau Basin, and the parallel chains of volcanic seamounts running from north to south along the Tonga Ridge. These oceanic features can potentially affect environmental parameters, like surface water temperature, nutrient levels, salinity, and dissolved oxygen concentration, all of which constitute the habitat conditions for pelagic species, like tuna. This could also explain why the four tuna species are consistently found in this area throughout the year, despite the fact that Tonga's EEZ is part of the Western and Central Pacific Ocean (WCPO) region, which is typically described as oligotrophic [69] due to extensive vertical mixing of water masses. Studies have demonstrated a strong correlation between tuna and shallow waters, particularly continental shelves and seamounts [70,71]. These locations are widely recognized as prime habitats for large offshore fishes, primarily because of the substantial foraging advantages they offer [71] and possibly for reproductive and navigational benefits [72–74].

The catch trends from 2002 to 2018 align with the patterns observed in the SOI during the same timeframe (Figure 7). Studies have shown that ENSO events are known to cause climate variability and are the major phenomena driving seasonal and inter-annual ocean processes in the Western Pacific [75]. ENSO events affect tuna catchability through the spatial shift of tuna's preferred habitat away from normal fishing grounds [76–78]. Other studies [46,79,80] have used species distribution modelling to predict the current and future distribution of tuna in relation to climate change. These studies have shown spatial and temporal shifts in their tuna abundance due to biophysical changes in their habitats. For example, Senina et al. (2018) [81] showed an eastern shift in the biomass of *skipjack* and *yellowfin* tuna over time at the Pacific Basin scale and within the EEZs of PICTs using the application of the model SEAPODYM applied for each tuna species. Considering the potential impacts of climate change on tuna distribution [46,82], knowledge of how environmental conditions influence CPUE becomes even more critical for Tonga. As changing ocean temperatures and currents can shift tuna habitats [83–85], understanding these relationships helps Tonga to anticipate and adapt to fluctuations in tuna availability, reducing economic and food security risks.

## 6. Conclusions

The significance of tuna species to the Pacific region, particularly to SIDs, like Tonga, cannot be overstated. Tonga relies on tuna fisheries to help sustain nutritional needs, provides opportunities for recreation, to contribute to government revenues, generate employment opportunities, support the welfare of their populations, and enrich their cultural heritage. Recognizing how critically important it is to safeguard these benefits, Tonga has prioritized the management of its tuna fisheries as a central development goal. This commitment is manifest through a dedicated effort to implement and uphold existing tuna management policies. Tonga's unwavering dedication is evident in its comprehensive approach, encompassing the vigilant monitoring of catch regulations and stringent oversight of foreign vessels operating within Tonga's EEZ. Central to this endeavor is the adoption of an information-driven management framework, exemplified by the TMDP, a dynamic strategy revised every five years to adapt to evolving challenges and opportunities.

Furthermore, we spatio-temporally merged the CPUE of four tuna species by longline fisheries in the EEZ of Tonga with environmental data from satellites and identified areas of high catch encounters by longline vessels. In terms of spatial distribution, the CPUE distribution pattern indicates a tendency for tuna species to be more concentrated in the central region of the EEZ. These habitat locations coincided with a period of high tuna encounters by longline vessels from April to October and low encounters between November and March. Regarding the distribution of environmental factors, it is evident from our monthly maps that sea surface current, sea surface salinity, and chlorophyll concentration exhibit consistent conditions throughout the year, without significant seasonal variations. For sea surface temperature, the distribution is not homogeneous in the EEZ of Tonga. Sea surface temperature displays noticeable seasonal fluctuations, with cooler temperatures occurring during the winter months from April to October, and warmer temperatures in the summer months from November to March, reaching around 30 $^{\circ}$C. There was, generally, a higher Chl-*a* concentration shown in the central part of the EEZ. Areas of moderate to strong El Nino events during the time span 2002 to 2018 correspond to periods of low catches of tuna in Tonga. We also show that periods of diminished catches in Tonga align with intervals of moderate to strong El Nino occurrences between 2002 and 2018. The study exclusively focuses on surface environmental factors to illustrate the significance of mapping these variables in relation to tuna catches. In the future, incorporating additional surface and subsurface environmental variables (such as sea surface height, mixed layer depth, dissolved oxygen levels, and water depth) will be helpful for detailed characterization of tuna habitats in Tonga's waters. Advancements, like employing species distribution modeling, will enhance our ability to gain a more accurate comprehension of the elements that impact the presence of tuna in Tonga, thereby contributing to the promotion of sustainable fisheries management and conservation initiatives.

**Supplementary Materials:** The following supporting information can be downloaded at: https://www.mdpi.com/article/10.3390/d15101042/s1. PDF file of the Tonga National Tuna Fishery Management and Development Plan 2018–2022 [6]. Figure S1. Spatio-temporal pattern in the distribution of *bigeye* tuna in the EEZ, longitude 14.5$^{\circ}$ S–20.22$^{\circ}$ S, longitude 171.31$^{\circ}$ W–179.10$^{\circ}$ W, of Tonga over the time period 2002–2018. Figure S2. Spatio-temporal pattern in the distribution of *skipjack* tuna in the EEZ, longitude 14.5$^{\circ}$ S–20.22$^{\circ}$ S, longitude 171.31$^{\circ}$ W–179.10$^{\circ}$ W, of Tonga over the time period 2002–2018. Figure S3. Spatio-temporal pattern in the distribution of *yellowfin* tuna in the EEZ, longitude 14.5$^{\circ}$ S–20.22$^{\circ}$ S, longitude 171.31$^{\circ}$ W–179.10$^{\circ}$ W, of Tonga over the time period 2002–2018.

**Author Contributions:** S.V. wrote the draft manuscript with input from S.K. All authors contributed to designing the study, the analysis, the interpretation of the results, the critical revision, and approval of the final manuscript. S.K. supervised the project. All authors have read and agreed to the published version of the manuscript.

**Funding:** The Pacific European Marine Program (PEUMP) grant number F3288 funded this research, and the APC. PEUMP is a multilateral fund financed by the European Union (EU) and the Swedish

International Development Cooperation Agency (Sida) aiming at building sustainable fisheries in the Pacific region.

**Institutional Review Board Statement:** Not applicable.

**Data Availability Statement:** All species data and extracted predictor variables are available in: DOI https://doi.org/10.5061/dryad.nk98sf7xs (accessed on 23 July 2023).

**Acknowledgments:** The authors would like to thank the Pacific European Marine Program for financially supporting this study and the Tonga Ministry of Agriculture, Forestry and Fisheries and the SPC for providing the fishery datasets from the Tonga longline fishery industry. Also, the authors acknowledge NOAA of the USA for the environmental and bathymetry remotely sensed datasets and the R community.

**Conflicts of Interest:** The authors declare no conflict of interest.

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
