# Peer review of "Ecosystem Management Policy Implications Based on Tonga Main Tuna Species Catch Data 2002–2018"

_diversity, doi:10.3390/d15101042_

Round 1

Reviewer 1 Report

General Comments

Climate change and its impacts on tuna fisheries have become one of the hot topics in tuna fisheries management over the last few years. There has been a significant uptick in research around the impacts of climate change, in particular on Pacific Island Communities. Further, the recent adoption of conservation and management measures in WCPFC, IOTC, ICCAT, and recently (a week back) in IATTC, shows that climate change is central in decision-making processes within RFMOs. The manuscript focuses on the importance of integrating climate sciences or indicators in decision-making, even from a national government. Thus, this contribution is very timely and valuable from a practical and theoretical perspective. 

However, the authors need to modify the manuscript substantially.  

1.     The authors need to have a clear structure and consistent flow throughout the paper rather than different ideas and different data at various sections. For example, the authors could follow a structure such as: an introduction including the questions asked in this research, the Tongan tuna fisheries and the management strategy, methodology including the data used and limitations (the use of climate and CPUE indices); analysis and results; discussion and conclusion.

The catches caught in Tongan EEZ are part of WCPO stock and tuna being a highly migratory species one country alone cannot manage the fishery. Thus, in the introduction, it is important to explain tuna management from a regional fishery (WCPFC) to the local perspective is essential. Further, in the WCPO, there are three subregional groups, the FFA, the PNA, and the SPG, which have different interests and often present different views in the WCPFC Commission meeting (which is the main body responsible for the management of tuna species). Bailey et al (2013), used a double agent problem to illustrate the relationship between RFMO and national fisheries. So, explaining the importance of protecting national interests in a regional multilateral body (WCPFC) presented by the authors using a similar framework might be helpful. It is also confusing that in section 3, the authors highlighted that Tonga supports ICCAT management when Tonga is not a member of ICCAT nor a coastal State in the management area.  

2.     It is difficult to understand the authors' use of skipjack tuna as the main species of analysis while albacore is the most abundant and most targeted species in the longline fishery in Tongan waters.

3.     Please use the scientific names of the species when the authors use the species name first.

4.     Try to use figures to explain the research question. If the authors have developed multispecies figures, use them as supplementary information. Also, while using the figures, explain the most important result from the figure.

5.     The authors need to modify the references, and the context of those references to the current reality of WCPFC. Some of the references are outdated and do not reflect the current reality of fisheries management in WCPFC.

6.     The conclusion must be revised based on the central message of the paper.

Detailed comments:

Abstract:

Line 12: Use Small Island Developing States (SIDS) rather than Small Island Countries

Line 17-18: The goal of Tonga National Tuna Fishery Management and Development Plan (2018 – 2022) includes "manage tuna resources through an ecosystem-based, precautionary and rights-based approach in order to maximise the benefits to Tonga's economy and people while ensuring the biological and economic sustainability of the fishery". The authors only focus on optimising benefits, rather than biological and economic sustainability. 

Line 20: The sentence needs to be more clear. Later in the manuscript, the authors discuss Tonga's work on monitoring catches to adhere to ICCAT and WCPFC resolutions. I suggest rephrasing to make the intent more straightforward. 

Line 22 – 28: This is the central aspect of this research, and the outcomes or the result of the analysis are missing. I suggest rephrasing. 

Introduction:

Line 48: The authors' citation (3) is from 2013, when the stock in WCPO was not in great shape. However, according to the latest SPC report presented in the WCPFC in 2022, all tropical tuna in WCPFC is healthy. See for further info: https://meetings.wcpfc.int/node/18208

Thus, the context of the paragraph might need to be more fitting for the current context. However, I agree that there are conflicting interests and issues concerning tuna management in WCPO.

Line 55: FFA is not a regional fisheries management organization. FFA is a regional body to help galvanize support work as a collective in fisheries management among the Pacific countries.

Line 60- 62: Since tuna is a highly migratory species, a nation alone cannot manage these species. Thus, under the UN Fish Stocks Agreement, to which Tonga is a party, countries must cooperate and collaborate in managing these species.

Line 65 – 68. How was the figure developed? Citations in Figure 1 show that it was summarised and modified from Bell and Tonga's TMDP. Can the authors explain how it was derived and modified? Can the authors also describe the figure and why it is needed in the paper? 

Section 2

Line 86 – 89: As explained earlier, this statement is invalid. Ecosystem impacts need to be taken into account even if the stocks are in a healthy state. The growth of the purse seiners has negative impacts on juveniles and, as a result, will impact member States who fish on large pelagic species such as Tonga. 

Line 91 – 113. I feel the authors could summarise the discussion about TMDP and the implementation schedule without going into extensive commentary. Some of the information is unnecessary.

Section 3:

Line 123 onwards: Authors mention that Tonga supports ICCAT CMMs. Can the authors explain the relationship between ICCAT and Tonga? Why does Tonga have to report catch data when it is not a member of ICCAT nor in the geographical scope of ICCAT?

Table 1: What is the reason behind choosing the data until 2018? 

Line 142: As mentioned earlier, this reference is outdated and should use the latest reference for the data.

Line 161 – 164: The references in the statements are very old and should be changed to use the current data, particularly the change in revenues after the implementation and growth of the VDS scheme.

Line 163- 164: Reference is missing.

Line 165: The phrase "given" could be replaced by "taken" or "caught." 

Line 197: The VDS is managed by the FFA, not WCPFC. 

Line 205 – 207: The FFA has a regional VMS and monitoring facility. Do the authors feel that it is inadequate? If so, could the authors elaborate more on the specific aspects of monitoring issues in the Pacific?

Line 214: The authors mentioned that Tong was the only FFA member without agreements with any DWFN. Can the authors explain the reasons behind it? 

Section 4:

Line 247 – 274: Can the authors explain the reasons behind the use of data till 2018? Further, there are too many details about the figure. It is essential to present the most important finding from the following figures. Some of the details could be moved to a methodology section.

Figures: It is evident that the most abundant species in Tonga is albacore, and the other figures could be used as supplementary information. It is important to keep the reader focused on the key findings and takeaways from the manuscript.

Section 4.2: Finally, the authors touch upon the methodology section here. It is essential to keep a flow in the paper rather than going in different directions. I've suggested an outline in general comments.

I still do not understand the rationale behind choosing skipjack as the main analytical species here. Longline fishing vessels target albacore, adult bigeye and adult yellowfin. Skipjack is often caught as a bycatch species.

Generally it was good. I have suggested few changes to the document to improve the flow and the english.

Author Response

Notes to reviewers is attached

Reviewer 2 Report

This is a historical overview of the tuna fishery in a small area of the South Pacific and of attempts to regulate its fishery for the sustainable exploitation of resources and the prevention of overfishing. This manuscript “also identified key research priorities that could be basis for national planning and policy development for conservation and sustainable management of tuna” (Ln 75-77). The last statement is not very specific: could or could not be; the policy for conservation and sustainable management of tuna stocks exists or there are only plans to develop one.

The area is considered in isolation from neighboring water areas without evidence that independent populations live there. An individual fish stock should be managed as a whole throughout the range, otherwise the regulation of the fishery will be ineffective. In addition, the article does not define the biological guidelines for stock management (for example, Flow, Fmed, Fhigh, Floss, F0,1 and Fmах).

The authors write about the need for an ecosystem-based approach to tuna management. However, it seems that not only an ecosystem, but even a population model has not been made for any of the species. Moreover, initial data, even for the simplest models, have not been collected - there are no data on the age composition of populations, the size of the spawning stock (SSB), recruitment, or mortality.

The Conclusion says: “Two of these regulations are ensuring tuna catch does not exceed sustainable levels (i.e., not more than its annual quotas) and obtaining all foreign vessels access and license fees” (Ln 459-461). But it does not say what these sustainable levels are for each species and how they are calculated.

Thus, this review is very superficial. While it may be useful to historians and fisheries lawyers, it is almost useless to basic research ichthyologists, applied fisheries science practitioners, and fisheries management officials.

Figure 1 is absolutely not informative. It may be removed from the manuscript without compromising its comprehension.

Figures 3-6 are not clear because there are too many images that are too small.

On Ln 420 “[67,71.72]” should be corrected to “[67,71,72]”.

I did not find a reference to the publication [75] (Ln 658-661) in the text of the manuscript.

The additional diversity-2568638-non-published.pdf file should not be in the form of scanned pictures of text, but in the form of source text, where you can select and copy words and phrases.

Author Response

Notes to reviewer is attached.

Reviewer 3 Report

1.      In general, this paper is well written and researched with new insight into current literature. I believe this paper can in some way fill up the gap between real practice and theory.

2.      At the end of Section 1, Introduction, there is a need to provide a brief paragraph concerning the overall structure of this paper.

3.      Between Section 3 and sub-section 3.1, you need to provide a brief introduction regarding the content of this section. Just like what you did in Section 4. This is for the consistency of overall presentation.

4.      “The most pressing management question” is teased out at the beginning of this paper but did not reflect at the end of the paper. I would suggest in the conclusion, you might want to in some way re-emphasize the pressing management question and provide answer to that question.  

Author Response

Notes to reviewer is attached.

Round 2

Reviewer 1 Report

General comments

I thank the authors for submitting a revised paper with a clear structure. However, with the new structure, the deficiencies highlighted in the 1st review are much more visible, and the authors must improve the paper.

I would like the authors to reflect on the questions in line 145 – 150 in this revised manuscript. The authors have answered the 1st question quite clearly, i.e., the review of Tongan tuna fisheries management strategies. However, the second part of the research is missing. There is a bit of a reflection in the conclusion, which has no link to the body of the text.

In the 1st review report, I highlighted the need to change the focus of the analysis from skipjack to albacore (the main target species in Tonga’s longline fisheries). However, the authors have not reflected this aspect in the reported findings other than just inserting a figure for the albacore fisheries and climate variables (Figure 5).

The methods section (section 3) and results (section 4) overlap significantly. The authors have justified the use of some of the indices in the result section (section 4) rather than in the methods. A methodology section should include more than just the data used but also a justification of the data or the indices used, which, unfortunately, is presented in the results section. I would like to advise the authors to be diligent in addressing the review comments. 

Figures must contribute to the manuscript and the flow of the text. Some of the figures do not add much value to the text, and the authors need to justify the use of it in the manuscript. 

Specific comments:

Abstract: What is the result of this paper? Is there something that the authors discovered during the analysis? The authors still need to address the comments in the first review fully. Please revise accordingly. 

Introduction:

Line 77 – 80: The authors have not corrected the 1st review comment: FFA is not a Regional Fisheries Management Organization. It is an advisory body. Please revise. 

Line 87 -96: Replace the word it with WCPFC. 

Line 84 – 119: There is significant duplication of the content in the two paragraphs. Please revise accordingly. 

Line 121- 123: One nation alone cannot manage highly migratory species. It was one of the major comments in the first review.

Figure 1: The authors have not justified the reasons for using Figure 1 in the text. I have raised the question in the first review, and the authors should include a few sentences to justify the use of this figure. 

Figure 3: The only inference from this figure was “The distribution of CPUE shown suggests that most of the fishing effort of the tuna longliners was concentrated in these geographical ranges”. What was the reason for the authors to show 2002 – 2018? Is there any other inference from the figure? For example, there is a change in fishing patterns compared to 2002 – 2003 to the most recent years. The CPUE towards the end of the year was highly concentrated in the NW, compared to the years from 2016 to 2018. Is it something significant? 

Figure 4. The figure could be moved into the result section as an introduction to the results. The figure could be improved to highlight a key finding from the research. For example, if there was a significant change in one species distribution over the years or whether a specific species is caught more in a particular month. When lumped together, the results will be skewed to the highest caught species, such as albacore and yellowfin tuna. 

Sections 3 and 4 need to be revised. 

No significant comment.

Author Response

Revisions is attached here.
